# Effects of Rhythm Step Training on Physical and Cognitive Functions in Adolescents: A Prospective Randomized Controlled Trial

**DOI:** 10.3390/healthcare10040712

**Published:** 2022-04-12

**Authors:** Sang-Kyun Park, Yong-Seok Jee

**Affiliations:** 1Department of Physical Education, Chungnam National University, Daehak-ro, Yuseong-gu, Daejeon 34134, Korea; ttl0033@gmail.com; 2Department of Leisure and Marine Sports, Hanseo University, Hanseo 1-ro, Haemi-myeon, Seosan 31962, Korea

**Keywords:** adolescents, physical function, cognitive function, rhythm step training

## Abstract

**Background:** Rhythm step training (RST) for sensorimotor dual tasks is in the spotlight as it provides physical activity that is fun and allows participants to express various and creative movements, although it lacks a scientific evidence base. Therefore, this study was to investigate how RST affects the physical and cognitive functions of adolescents. **Materials and Methods:** A total of sixty-six female middle-schoolers were divided into non-exercise group (control group, CON, *n* = 22), step training group (STG, *n* = 22), and rhythm step training group (RSTG, *n* = 22). To verify the combined effects of music-based rhythm and exercise, the program was conducted for 45 min/session a day, three times a week for 12 weeks. **Results:** RST scores increased significantly in the STG and RSTG compared to the CON after 12 weeks. Specifically, the Δ% of RST scores in the RSTG (11.44%) was higher than those of STG (9.01%) and CON (3.91%). By the end of the experiment, the power (*p* < 0.001), agility (*p* < 0.001), muscle endurance (*p* < 0.001), dynamic or static balance (*p* < 0.001), and gait velocity (*p* < 0.001) of RSTG were significantly improved compared to the others. The Δ% of all variables in RSTG was higher than those of the CON or STG. In addition, the verbal memory (*p* < 0.001) and attention (*p* < 0.001) of cognitive function were significantly improved in RSTG. Specifically, there was more of an increase in Δ% of RSTG for verbal memory (7.52%) and attention (10.33%) than in the CON (verbal memory, 3.34%; attention, 5.83%) or STG (verbal memory, 5.85%; attention, 5.43%). **Conclusions**: This study confirms that RST had a positive effect on the physical and cognitive functions of female middle-schoolers. We propose that rhythmic exercise combined with music is beneficial for adolescents’ physical and cognitive health.

## 1. Introduction

Life expectancy is increasing, in part due to the development of medical technology, although there are many other reasons behind this increase. Physical inactivity related to life expectancy is 35% in Korea due to poor lifestyle habits, so it is recommended that people increase their rate of physical activity for a healthier life [1]. The coronavirus-19 (COVID-19) pandemic continues around the world, and measures to restrict its spread have led to unhealthy or irregular exercise patterns and increased physical inactivity. According to one study, it was reported that adolescents are negatively affected not only in school life, but also in social life and outdoor activities due to COVID-19 social distancing measures. Some of them have experienced an increase in domestic violence and report that similar events are directly affecting their psychological health, such as stress, anxiety, and fear [2]. In addition, the decrease in social activities, such as restrictions on the use of sports facilities and school attendance due to COVID-19, is emerging as a big problem for adolescents in their growth stages [3,4]. As a result of this, physical activity at school has become more difficult than in the past. In the COVID-19 era, regular exercise or physical activity is essential as it helps to improve our body and mind [5,6,7]. Above all, exercise is especially important for adolescents who are in their growth stages [8]. It is reported that regular exercise for adolescents not only improves their physical health, but also helps to develop their cognitive function [9].

In Korea, most schools have been reducing physical education hours while emphasizing the importance of other major subjects. As a result, adolescents have become accustomed to this situation and have neglected physical activity classes. The COVID-19 pandemic has further reduced the amount of time requiring physical activity [10]. It is necessary to develop a strategy that not only increases the time dedicated to adolescents’ physical activity, but also increases their likelihood of participating in regular physical activity.

Many studies have suggested that exercise combined with music can induce interest [11] and is suitable for improving physical and cognitive functions [12,13]. It is also reported that rhythmic exercise improved not only the health-related physical fitness level of children and adults with intellectual disabilities [14,15], but also mobility and fear of falling in the elderly [16]. Such a program that mixes physical activity or movement with elements of other sensory domains is called a ‘dual task’. Based on our experiences in the field of physical education classes, we consider that adding music is often more effective in various regards than physical activities performed without accompaniment. In this context, several researchers have reported that sensorimotor training of dual-task programs is effective for improving physical conditioning, as well as cognitive function [17,18].

The recently developed rhythm step training (RST) for sensorimotor dual-task training for adolescents has gained much attention as it provides physical activity with an element of fun and allows them to express various and creative thoughts during the active movement process. Previous attempts at step training without music or rhythm might have bored the adolescents as they involved repeating simple patterns without enjoyment. Improving on this, the step ladder training of RST enables the easy adjustment of the level of difficulty by changing the movement patterns. Its frequency of use is increasing day by day due to people’s favorable impression of this approach, the small space required, and the low cost. RST is an exercise program that adds upper-extremity movements to the pre-existing step training that focuses on lower-extremity movements, and it incorporates music as a continuing participation strategy [19]. The basic motions consist of a jumping step with one foot, a jumping step with both feet, a sitting and standing jump step, a lower-extremity raising motion, an upper-extremity motion, and a motion where a hand touches a body part. This configuration can raise the interest of participants with various motion patterns and encourage them to engage in whole-body exercise. Sitting up and raising the lower extremities in RST, rather than being simple motions, have the effect of strengthening the muscles around the ankle, knee, and hip joints. In addition, it is effective in motivating participants as they can select an RST program that suits them, and it is possible to increase participation in exercise without facing time or spatial restrictions. However, no one has yet studied how helpful RST is for raising adolescents’ physical fitness level and how much it influences their cognitive function. Therefore, this study aims to investigate the effect of RST on adolescents’ physical and cognitive functions.

## 2. Materials and Methods

### 2.1. Study Design and Participants’ Characteristics

This study was designed as a prospective, randomized, controlled trial to verify the effects of RST on adolescents’ physical and cognitive functions. This study measured the physical function based on power, agility, muscle endurance, static or dynamic balance, and gait ability. In addition, we measured verbal memory and attention to determine cognitive function. Before the study, the principal investigator explained all procedures to the participants and their parents or legal guardians who then read and signed a consent form. This study was conducted according to the Declaration of Helsinki (2013 version) and was approved by the Institutional Review Board of Chungnam National University (202001-SB-008-01). The 12-week experiment sought to observe the physical and cognitive functions of adolescents.

The adolescents were female middle-schoolers from Daejeon, Korea. Their ages ranged from 13–16 years old. The inclusion criteria for this study were (1) voluntary participation in RST, (2) lack of regular exercise over the previous six months, and (3) no specific disease. The exclusion criteria were: having joint problems; a history of taking medication; undergoing surgery in the past six months; having depression; other mental health problems; or an intellectual disability. Initially, eighty-two girls enrolled in the study. All participants were assigned using random number tables and identification numbers on recruitment to the study. However, four adolescents were excluded due to severe obesity or having undergone a recent operation. Thus, seventy-eight participants were randomly assigned to three groups of twenty-six people each, as shown in Figure 1.

As shown in Figure 1, before entering the analysis phase, four participants from the control group (CON), four participants from the step training group (STG), and four participants from the rhythm step training group (RSTG) were lost. Finally, a total of sixty-six participants were included in the study. The participants in the CON engaged neither in the step training nor in the rhythm step training. To prevent communication between the groups, the participants were classified according to their area of residence. The interventions and measurements were also scheduled at different times as follows: CON at 16:00, STG at 17:00, and RSTG at 18:00.

### 2.2. Measurement Methods

#### 2.2.1. RST Measure

The RST ability was assessed using an RST evaluation scale, as shown in Table 1. The draft RST scale, paired with a 0–4-point Likert scale, was created through consultation and discussion with three experts and based on the results of a preliminary test conducted with sixty adolescents who participated in a four-week RST program. The fundamental movement skills (FMS) are mainly divided into object control, locomotor skills, and stability skills. FMS assessments are applied or modified to observe movement skills’ development and mastery [20,21,22]. We employed the Canadian Agility and Movement Skill Assessment [22], which assesses the completion time, time score, skill score, and total score, and the repeated sidestep test [20,23], which measures only the number of times a pattern is performed accurately, without inaccurate steps. Thus, in this study, the RST scale came to be composed of the time score (temporal variables), accuracy score (spatial variables), and total score. In the RST ability test, the participants performed five-step patterns, once each, on two stair ladders at a distance of 10 m, for a total of five times, and the temporal and spatial scores were summed to give a total score. For the temporal variables, the total execution time was recorded based on the RST temporal scale. For the spatial variables, the number of times that a participant failed to step on the space determined according to the pattern was recorded based on the RST spatial scale. When calculating Cronbach’s-α, the reliabilities for the temporal and spatial variables were *r* = 0.778 and *r* = 0.885, respectively, demonstrating internal consistency, and the inter-scorer reliability was *r* = 0.997.

#### 2.2.2. Physical Function Measures

The physical functions were composed of power (one-leg hop test) [24], agility (carioca test) [25], muscle endurance (crossover step-up test) [26,27], dynamic balance ability (tandem walk test) [28], and static balance ability (star excursion balance test, SEBT) [29]. First, the one-leg hop test started with jumping on one leg and measured the distance jumped with three consecutive jumping motions. The maximum value of two measurements was recorded. Second, the crossover step-up test recorded the number of leg positions exchanged for 1 min by placing one leg on a 20-cm step box and passing the leg to the opposite side of the step box. After both legs were alternately measured once, the mean value was calculated from the values from both legs. Third, the carioca test measured the time taken to move a 10-m distance as a sidestep by alternating the feet and moving both feet in one direction, alternately back and forth. After both directions were alternately measured once, the mean value was calculated from the values for both directions. Fourth, the tandem walk test measured the time taken to walk a distance of 10 m on a straight line without losing their balance by placing down the heel and forefoot alternately. If a subject missed the line or walked in a straight line without the heel and toe touching, we repeated the measure. The maximum value of two measurements was recorded. Fifth, the SEBT measured the distance where the participants could support their weight on one foot and stretch the other leg as far as possible along lines in eight directions, reaching away with the opposite foot. The range of distance measurement was from the center to the tip of the big toe of the extended leg. After both legs were alternately measured once, the mean value was calculated from the values for both legs. Last, the gait ability was measured using a gait analyzer (LegSys+, Newton, MA, USA, 2016), recording the temporal (velocity and cadence) and spatial gait characteristics (stride length). The portable sensors of the equipment were worn on the participants’ ankles, and they walked a distance of 10 m. During the measurement, participants were asked to walk at their natural gait speed.

#### 2.2.3. Cognitive Function Measures

We assessed the cognitive function based on the verbal memory (word list memory test) and attention (trail making test) of the adolescents [30]. First, in a word list memory test, we performed a free recall task using 10 commonly used nouns to evaluate participants’ ability to remember newly learned information. After 10 words were presented at a constant pace, the participants responded by recalling as many words as possible in 90 s. The maximum value of two measurements was recorded. Second, the trail-making test was to connect 25 randomly arranged circles of numbers in order with pencil lines as quickly as possible, and the total time required was recorded in seconds.

### 2.3. RST Program

The RST program was performed for 45 min a day, three days a week for 12 weeks, as shown in Table 2. The program is performed according to a pattern by moving to a space divided into a line or a step ladder.

It has the following differentiation compared to the pre-existing step training: First, it is possible to increase participants’ attention and interest through rhythm change, motion transformation, and motion creation, as well as by listening to preferred music. Second, 20 basic patterns can be applied to line deformation in three stages and rhythm motion deformation in five stages. The line deformation consisted of three steps (one line step, one step-ladder step, and two step-ladder steps). The rhythmic motion deformation consisted of five steps (normal step, split step, normal step with arm motion, split step with arm motion, and high speed with creative motion). Last, the goals to be achieved can be set step-by-step, and achievement levels can be identified by a measurement scale, which can induce motivation. The intensity of all exercises was measured by ratings of perceived exertion (RPE) [31]. For reference, the step training performed by the STG is an exercise that has been generally used in the past. It is to move the agility ladder (step ladder) quickly and repeatedly according to a set pattern without music.

### 2.4. Data Process and Statistical Analyses

The experiment design of this study was to analyze the group, time, and interaction by repeated measures of analysis of variance (ANOVA). Using the G*power program (Heinrich-Heine-University Software, Duesseldorf, Germany) [32], 66 subjects were analyzed when group = 3 and time = 2 were substituted on the basis of effect size = 0.25, α = 0.05, and β = 0.95. Effect size (η^2^) was calculated according to Cohen’s *d*, which is equal to the mean difference in the groups divided by the pooled SD [33]. Microsoft Excel (Microsoft, Redmond, WA, USA) was used to organize the data. All data measured were analyzed by SPSS^®^ 24.0 (IBM Corporation; Armonk, NY, USA). The reliability of the RST scale was calculated as the internal consistency and inter-scorer reliability through Cronbach’s α. The normality of the study was calculated by the Shapiro–Wilk test. After primary analysis with repeated measures of ANOVA, the paired *t*-test was conducted to investigate the changes between pre- and post-values in each group, and Tukey’s post hoc test was implemented if there were significant differences in the time, group, and group-by-time interaction. Finally, the one-way ANOVA using GraphPad Prism 9.3.1 (La Jolla, CA, USA) for delta percentage (Δ%) was conducted to compare the changes between times. The Δ% was calculated using the formula of {(postdata–predata)/predata} × 100. For all analyses, the significance level was set at *p* ≤ 0.05.

## 3. Results

### 3.1. Comparisons of Demographic Data

As shown in Table 3, the body weights of CON, STG, and RSTG were not significantly different. The percentage of fat and body mass index (BMI) were also not significantly different. These results showed that there was homogeneity before the experiment.

### 3.2. Difference and Change in RST Ability

As shown in Table 4, the rhythm step training score significantly increased in the STG and RSTG after 12 weeks, although there was no change in the CON. As shown in Figure 2, the Δ% of RSTG (11.44%) was significantly higher than those of CON (3.91%) and STG (9.01%).

### 3.3. Effect of RST on Physical Functions

As shown in Table 5, there were significant differences in the time and group-by-time interaction of power or one leg hop (pre η^2^ = 0.034; post η^2^ = 0.043), agility or carioca (pre η^2^ = 0.025; post η^2^ = 0.001), muscular endurance or cross over step up (pre η^2^ = 0.007; post η^2^ = 0.026), dynamic balance or Tandem walk (pre η^2^ = 0.056; post η^2^ = 0.129), cadence (pre η^2^ = 0.057; post η^2^ = 0.075), and velocity (pre η^2^ = 0.036; post η^2^ = 0.049). There were also significant differences in the time and group-by-time interaction of anterior SEBT (pre η^2^ = 0.020; post η^2^ = 0.045), anterolateral SEBT (pre η^2^ = 0.004; post η^2^ = 0.006), lateral SEBT (pre η^2^ = 0.001; post η^2^ = 0.012), posterolateral SEBT (pre η^2^ = 0.013; post η^2^ = 0.044), posterior SEBT (pre η^2^ = 0.018; post η^2^ = 0.035), posteromedial SEBT (pre η^2^ = 0.011; post η^2^ = 0.034), medial SEBT (pre η^2^ = 0.006; post η^2^ = 0.023), and anteromedial SEBT (pre η^2^ = 0.010; post η^2^ = 0.006).

Although the power of CON (0.63%) and STG (1.11%) increased, that of RSTG (2.36%) increased more than the other groups, as shown in Figure 3A. The agility of CON (−3.01%) and STG (−5.70%) decreased, however, that of RSTG (−13.34%) decreased more than the other groups, as shown in Figure 3B. In other words, it could be seen that the agility became faster. On the other hand, in the case of muscular endurance, CON (2.37%) and STG (3.62%) improved after 12 weeks, although the value (5.08%) of RSTG showed greater improvement. These variables were significantly different among the groups, as shown in Figure 3C.

Dynamic balance decreased by −1.52% in CON and −8.20% in STG, however decreased by −9.18% in RSTG, as shown in Figure 4A. Static balance showed various changes in the three groups, and as a result of averaging the total scores for eight tasks, it increased by 2.08% in CON and 3.44% in STG, whereas it increased by 4.60% in RSTG. As a result of the one-way analysis of variance, there was a statistically significant difference between the groups, as shown in Figure 4B.

After 12 weeks, the cadence of gait ability increased by 1.20% in CON and 0.61% in STG, however increased by 0.15% in RSTG, as shown in Figure 5A. Similarly, the velocity of gait ability increased by 1.15% in CON and 1.52% in STG, yet increased by 2.65% in RSTG. As a result of the one-way analysis of variance, there was a statistically significant difference between the groups, as shown in Figure 5B.

### 3.4. Effect of RST on Cognitive Function

As shown in Table 6, there were significant differences in the times taken for the word list memory test (verbal memory) and trail-making test (attention), and in the group-by-time interaction of verbal memory and attention.

After 12 weeks, the Δ% of verbal memory of cognitive function increased by 3.34% in CON and 5.85% in STG, however increased by 7.52% in RSTG. As a result of the one-way analysis of variance, there was a statistically significant difference between the groups, as shown in Figure 6A. Similarly, the Δ% of attention of cognitive function decreased by −5.83% in CON and −5.43% in STG, yet decreased by 10.33% in RSTG. As a result of the one-way analysis of variance, there was a statistically significant difference between the groups, as shown in Figure 6B.

## 4. Discussion

This study found that the RST ability, power, agility, muscle endurance, balance ability, and gait ability of the rhythm step training group all significantly improved after the 12-week experiment was completed. Furthermore, it was found that verbal memory and attention of the RSTG were significantly improved after 12 weeks. Rhythm step training is an exercise program that adds upper-extremity movements to the pre-existing step training that focuses on lower-extremity movements, and it incorporates music as a continuing participation strategy [19]. The basic motions consist of a jumping step with one foot, a jumping step with both feet, a sitting and standing jump step, a lower-extremity raising motion, an upper-extremity motion, and a motion where a hand touches a body part. This configuration can raise the interest of adolescents with various motion patterns and encourage them to engage in whole-body exercise. Sitting up and raising the lower extremities in rhythm step training, rather than being simple motions, have the effect of strengthening the muscles. In addition, it is effective at motivating adolescents as they can select a rhythm step training program that suits them, and it is possible to increase participation in exercise. The beneficial effects of rhythm step training can be seen through the improvements in the body function variables observed in this study.

Generally, power is increased through improved muscle strength. This is demonstrated when the muscle fiber momentarily contracts, based on the rate of contraction of the skeletal muscle groups [34]. Rhythm step training includes repetitive jumping motions, which mobilize the motor unit maximally. The increased skeletal muscle contraction through the force applied when jumping and landing improve muscle functions. Rhythm step training also includes a sit-jump step that lifts the lower extremities and a heel-up jump step, in addition to the general jumping motion. These jumping motions improve power and muscle endurance by increasing the repetitive movements of lower extremities, thus promoting neuromuscular activation and coordination [35,36]. In this context, Aloui et al. [37] reported that repetitive jump motions and various actions help improve agility by promoting the neuromuscular sense and physical function, as muscles repeatedly contract due to the interaction of several neuromuscular adaptations and improve the neural drive to agonist muscle groups.

This study also included the rhythm step training motions that require accuracy and coordination between upper and lower extremities. In addition, there were direction-changing motions such as parallel jumps and turn jumps and agility skills that require neuromuscular control and activation for proprioceptors. Through the rhythm step training of this experimental process, this study was able to assess the results of developing agility and muscular endurance, as shown in Figure 3. The rhythm step training based on repetitive lower extremities movements for 12 weeks improved participants’ static and dynamic balance, as shown in Figure 4. In this context, Aloui et al. [38] reported that the balance increases by enhancing the muscle strength of the lower extremities, improving the movement of the knee and hip joints, and strengthening the core muscles. The results of the study presented are further supported by those of previous studies that demonstrated exercise using jumps was effective at improving agility, muscle endurance, and balance by improving the proprioceptor functions and strengthening the skeletal muscles around the lower extremities [39,40]. Furthermore, this study found that there were significant increases in cadence and velocity after 12 weeks, as shown in Table 5 and Figure 5. In terms of cadence, the CON significantly increased compared to the RSTG and STG, although the cadence score of STG was significantly higher than that of the RSTG. These results indicate that, through 12 weeks of rhythm step training, the speed for the gait of RSTG increased, while at the same time, the number of steps per minute decreased relatively as stride length increased. According to You et al. [41], repetitive step training generates effective synaptic potentiation, thereby creating associated motor improvement. Leroux et al. [42] and Karabin et al. [43] suggested that efficient walking is possible by reducing the sway of the torso and upper extremities to maintain the center of gravity when walking. Another factor that improves velocity is increasing the stride length by minimizing postural sway [43]. In other words, the previous studies have reported that rhythm step training, which involves many movements of the center of the body, was shown to improve gait ability and balance through the performance of frequent, repetitive body movements, as do the results of this study.

Along with an improvement of physical functions, rhythm step training may improve cognitive function by applying upper- and lower-extremity coordinating motions and creating various patterns using rhythm changes. This study observed that after 12 weeks of rhythm step training, there were significant increases in verbal memory and attention, as shown in Table 6 and Figure 6. We think these results correspond with several papers in finding that an exercise improves cognitive function by influencing neurotrophic and nerve growth factors secreted from the hippocampus [44,45,46]. In addition, our findings are in agreement with research on how aerobic exercise positively affects brain functions such as hippocampal endothelial cell proliferation and vascular cell generation by improving cardiorespiratory fitness [47,48], as rhythm step training is an aerobic exercise in which body movement is the main activity. In this context, Hillman et al. [49] reported on a body of multidisciplinary literature that has documented the benefit of physical activity through aerobic exercise for selective aspects of brain function. They demonstrated that aerobic exercise can improve many aspects of cognition and performance. If all adolescents were to take part in rhythm step training, it might not only help to improve their physical health, but might also raise their academic performance. Adding to the existing literature, the work presented here examined and observed the positive effects of rhythm step training on cognitive and physical function at behavioral levels. However, this study has some limitations in that it is difficult to apply the findings to all age groups as the participants in this study were female adolescents. Furthermore, more extensive validation work with a larger sample size is needed in the future. Given these limitations, further studies investigating the effect on a more diverse and larger number of subjects are recommended.

## 5. Conclusions

This study found that the use of a 12-week RST program for female middle-school students improved their physical and cognitive functions. These results should raise awareness of the importance of exercise for adolescents who have difficulty participating in physical activity due to social distancing and the spread of a culture of refraining from outside activities brought on by the COVID-19 pandemic. Ultimately, rhythm step training can be used as a strategic tool to promote continuous voluntary participation in physical activities in school sports or social sports fields by utilizing appropriate music for the participants. Moreover, we found that physical activity combined with music is very important for adolescents as it improves their cognitive function.

## Figures and Tables

**Figure 1 healthcare-10-00712-f001:**
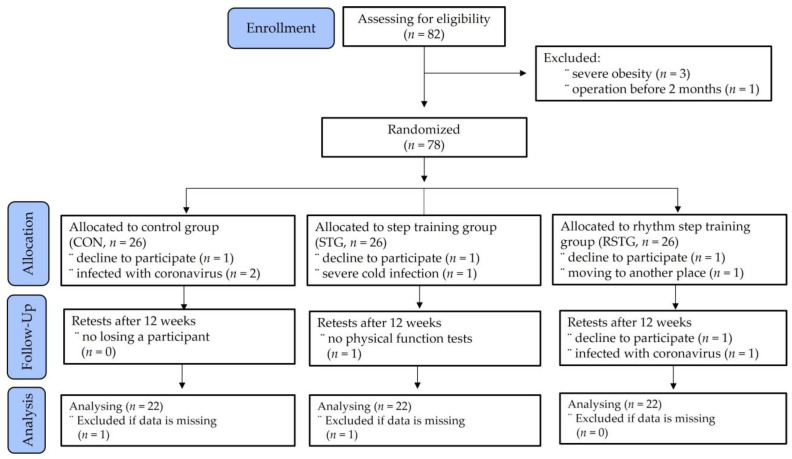
Participants’ allocation (consolidated standards for reporting of trials flow diagram).

**Figure 2 healthcare-10-00712-f002:**
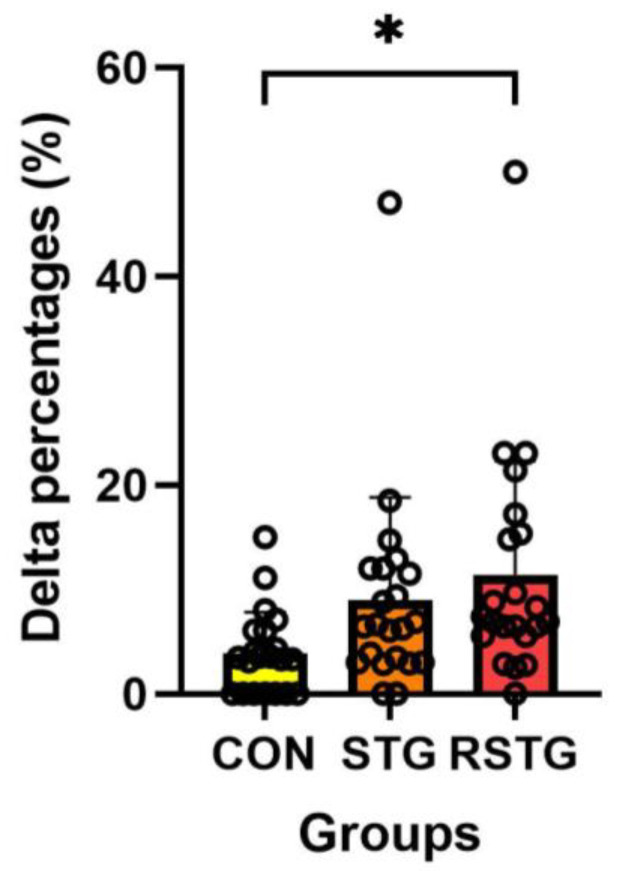
Difference in Δ% of RST ability among three groups. All data represent mean and standard deviation marked circles. CON, control group; STG, step-training group; RSTG, rhythm step training group. *****, *p* < 0.05.

**Figure 3 healthcare-10-00712-f003:**
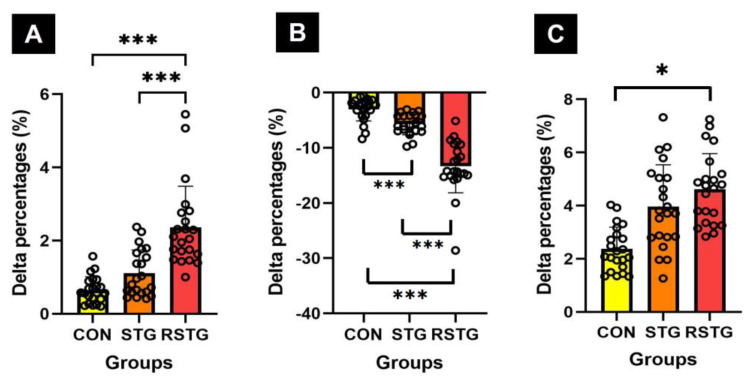
Difference in Δ% in power, agility, and muscular endurance. Here, (**A**–**C**) represent power, agility, and muscular endurance, respectively. All data represent mean and standard deviation marked circles. CON, control group; STG, step-training group; RSTG, rhythm step training group. *****, *p* < 0.05; *******, *p* < 0.001.

**Figure 4 healthcare-10-00712-f004:**
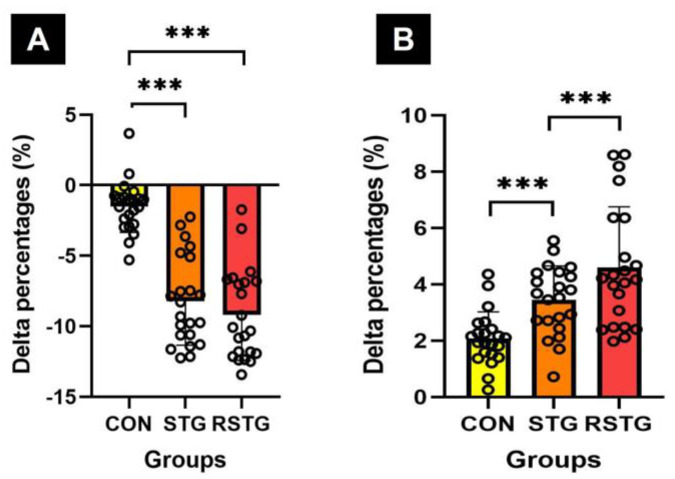
Differences in Δ% of dynamic and static balances. Here, (**A**,**B**) represent dynamic balance (**A**) and total mean static balance (**B**) scores, respectively. All data represent mean and standard deviation marked circles. CON, control group; STG, step-training group; RSTG, rhythm step training group. ***, *p* < 0.001.

**Figure 5 healthcare-10-00712-f005:**
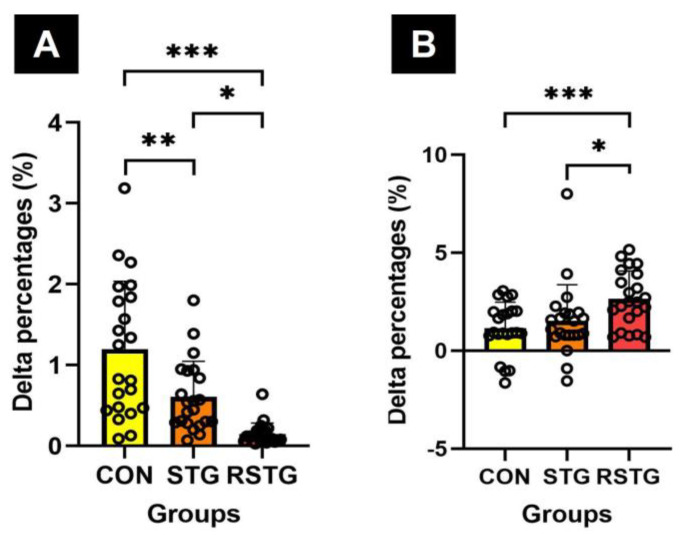
Difference in Δ% of cadence and velocity of gait ability. Here, (**A**,**B**) represent cadence (**A**) and velocity (**B**) scores, respectively. All data represent mean and standard deviation marked circles. CON, control group; STG, step-training group; RSTG, rhythm step training group. *, *p* < 0.05; **, *p* < 0.01; ***, *p* < 0.001.

**Figure 6 healthcare-10-00712-f006:**
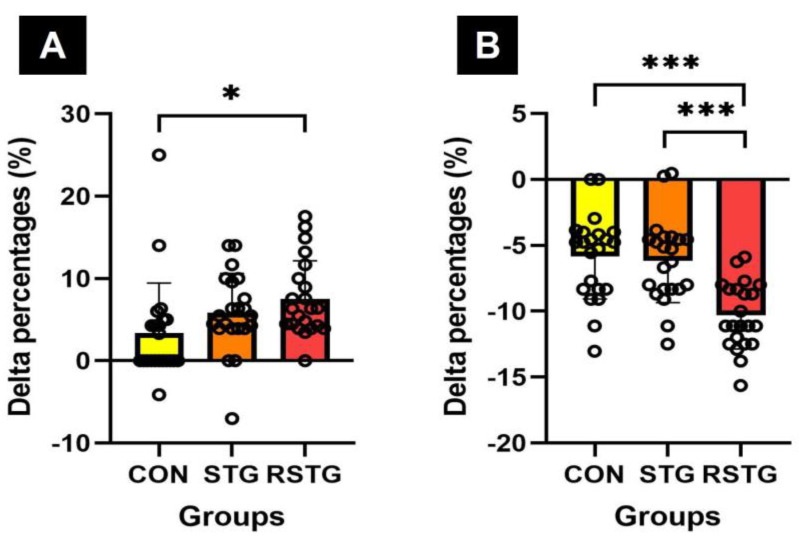
Difference in Δ% of verbal memory and attention of cognitive function. Here, (**A**,**B**) represent verbal memory (**A**) and attention (**B**) scores, respectively. All data represent mean and standard deviation marked circles. CON, control group; STG, step-training group; RSTG, rhythm step training group. *, *p* < 0.05; ***, *p* < 0.001.

**Table 1 healthcare-10-00712-t001:** Temporal and spatial measures’ scales for performance time and accuracy.

Items	Scale	Forward Step	Sidestep	One Leg Step	Side to Sidestep	Zigzag Step
Time (s)	0	>10.5	>11.75	>8.25	>11.5	>16.5
	1	9–10.5	9.75–11.75	7.25–8.25	9.5–11.5	14.5–16.5
	2	7.5–9	7.75–9.75	6.25–7.25	7.5–9.5	12.5–14.5
	3	6–7.5	5.75–7.75	5.25–6.25	5.5–7.5	10.5–12.5
	4	<6	<5.75	<5.25	<5.5	<10.5
Inaccuracy (reps.)	0	≥4	≥4	≥4	≥4	≥7
	1	3	3	3	3	5–6
	2	2	2	2	2	3–4
	3	1	1	1	1	1–2
	4	0	0	0	0	0

**Table 2 healthcare-10-00712-t002:** Rhythm step training program for RSTG.

	Exercise Types	ET	RPE
Warmup	Stretching for whole body	10 min	7–9
Workout	① Forward direction: jump step, two feet jump step ② Backward direction: jump step, two feet jump step ③ Side direction: jump step, two feet jump step, sit to jump ④ Hop and two feet jump step ⑤ Cross direction: jump step, two feet jump step ⑥ Parallel cross: jump step, two feet jump step ⑦ Parallel direction: jump step, two feet jump step ⑧ Turn 360°: jump step, two feet jump step ⑨ Side to side: jump step, two feet jump step ⑩ Heel-up and two feet jump step ⑪ Creative rhythm jump step	25 min	12–14
Cooldown	Stretching for whole body	10 min	7–9

ET, Exercise time; RPE, Ratings of perceived exertion.

**Table 3 healthcare-10-00712-t003:** Physical characteristics of the participants.

	Groups				
	CON (*n* = 22)	STG (*n* = 22)	RSTG (*n* = 22)	F	*p*
Age (y)	14.82 ± 0.80	14.68 ± 0.84	14.41 ± 0.96	2.652	0.078
Height (cm)	158.23 ± 5.44	158.55 ± 4.48	159.68 ± 4.29	0.567	0.570
Weight (kg)	60.36 ± 12.29	55.64 ± 9.60	54.73 ± 9.20	1.850	0.166
Percent fat (%)	24.14 ± 5.03	22.00 ± 3.52	21.45 ± 3.66	2.597	0.082
BMI (kg/m^2^)	20.60 ± 8.20	19.81 ± 5.92	23.31 ± 5.74	1.655	0.199

All data represent mean ± standard deviation. Comparative analysis was performed by one way ANOVA. BMI, body mass index; CON, control group; STG, step training group; RSTG, rhythm step training group.

**Table 4 healthcare-10-00712-t004:** Differences and changes in RST ability.

Time (T)		Groups (G)			F (*p*)			
		CON	STG	RSTG	T	G	G × T	η^2^
RST ability	pre	28.36 ± 4.34	29.59 ± 4.34	30.64 ± 5.94	114.506	2.705	7.570	0.036
	post	29.45 ± 4.58	32.00 ± 4.58 ***	33.59 ± 4.76 ***	(<0.001)	(0.075)	(<0.001)	0.136

All data represent mean ± standard deviation. CON, control group; STG, step-training group; RSTG, rhythm step training group; G, group; T, time; G × T, group × time interaction. *** means *p* < 0.001.

**Table 5 healthcare-10-00712-t005:** Differences and changes in physical function.

		Time (T)	Groups (G)			F (*p*)		
			CON	STG	RSTG	T	G	G × T
One leg hop (cm)		pre	413.41 ± 56.49	443.14 ± 56.49	432.32 ± 81.15	334.261 (<0.001)	1.255 (0.292)	45.920 (<0.001)
		post	416.00 ± 56.92 ***	447.82 ± 56.92 ***	441.95 ± 80.24 ***			
Carioca (s)		pre	14.15 ± 4.55	14.25 ± 4.55	15.60 ± 3.99	367.167 (<0.001)	0.217 (0.805)	71.845 (<0.001)
		post	13.72 ± 4.40 ***	13.46 ± 4.40 ***	13.61 ± 3.96 ***			
Cross over step up (rep/min)		pre	69.71 ± 7.77	69.59 ± 7.14	71.32 ± 12.71	365.150 (<0.001)	0.468 (0.628)	15.812 (<0.001)
		post	71.32 ± 7.63 ***	72.05 ± 6.85 ***	74.73 ± 12.36 ***			
Tandem walk (s)		pre	21.34 ± 6.05	18.56 ± 6.05	19.94 ± 4.65	279.514 (<0.001)	3.035 (0.055)	37.188 (<0.001)
		post	20.99 ± 5.86 *	17.01 ± 5.86 ***	18.10 ± 4.27 ***			
SEBT (cm)	Anterior	pre	57.18 ± 4.56	58.73 ± 4.56	57.86 ± 4.37	413.799 (<0.001)	0.980 (0.381)	19.934 (<0.001)
		post	58.07 ± 4.52 ***	60.25 ± 4.52 ***	59.86 ± 4.26 ***			
	Anterolateral	pre	58.32 ± 5.56	58.34 ± 5.56	57.64 ± 5.79	233.837 (<0.001)	0.031 (0.970)	16.975 (<0.001)
		post	59.23 ± 5.31 ***	59.91 ± 5.31 ***	60.07 ± 5.15 ***			
	Lateral	pre	58.07 ± 6.06	58.20 ± 6.06	57.89 ± 7.25	398.957 (<0.001)	0.087 (0.917)	13.335 (<0.001)
		post	59.27 ± 5.86 ***	60.45 ± 5.86 ***	60.64 ± 6.44 ***			
	Posterolateral	pre	60.80 ± 7.14	60.68 ± 7.14	62.32 ± 7.37	161.165 (<0.001)	0.820 (0.445)	9.601 (<0.001)
		post	61.75 ± 6.71 ***	62.43 ± 6.71 ***	64.70 ± 6.36 ***			
	Posterior	pre	63.86 ± 8.82	66.45 ± 8.82	64.64 ± 9.21	116.612 (<0.001)	0.775 (0.465)	6.648 (0.002)
		post	64.93 ± 7.98 ***	68.20 ± 7.98 ***	67.18 ± 8.22 ***			
	Posteromedial	pre	61.95 ± 8.24	64.20 ± 8.24	63.16 ± 11.16	87.397 (<0.001)	0.656 (0.522)	5.665 (0.005)
		post	62.91 ± 7.79 ***	66.20 ± 7.79 ***	65.70 ± 10.01 ***			
	Medial	pre	54.68 ± 8.12	56.23 ± 8.12	55.70 ± 9.83	179.408 (<0.001)	0.404 (0.670)	5.503 (0.006)
		post	56.18 ± 8.01 ***	58.75 ± 8.01 ***	58.52 ± 8.70 ***			
	Anteromedial	pre	53.59 ± 5.50	52.16 ± 5.50	52.50 ± 7.72	233.340 (<0.001)	0.218 (0.804)	4.303 (0.018)
		post	55.25 ± 5.26 ***	54.34 ± 5.26 ***	55.18 ± 6.87 ***			
Gait ability	Cadence (rep/min)	pre	113.20 ± 10.85	115.04 ± 5.50	109.95 ± 9.18	96.761 (<0.001)	2.195 (0.120)	20.885 (<0.001)
		post	114.52 ± 10.63 ***	115.74 ± 5.48 ***	110.11 ± 9.13			
	Velocity (m/sec)	pre	1.06 ± 0.15	1.14 ± 0.14	1.13 ± 0.22	98.975 (<0.001)	1.377 (0.260)	6.789 (0.002)
		post	1.07 ± 0.15 ***	1.15 ± 0.14 ***	1.15 ± 0.22 ***			
	Stride length (m)	pre	1.18 ± 0.16	1.18 ± 0.12	1.15 ± 0.14	1.189 (0.280)	0.055 (0.947)	0.469 (0.628)
		post	1.18 ± 0.10	1.20 ± 0.12	1.21 ± 0.15			

All data represent mean ± standard deviation. SEBT, star excursion balance test; CON, control group; STG, step-training group; RSTG, rhythm step training group. G, group; T, time; G × T, group × time interaction. As a result of paired *t*-test, if there is a significant difference between time, * means *p* < 0.05, and *** means *p* < 0.001.

**Table 6 healthcare-10-00712-t006:** Differences and changes in cognitive function.

	Time (T)	Groups (G)			F (*p*)			
		CON	STG	RSTG	T	G	G × T	η^2^
Verbal memory	pre	67.50 ± 12.03	65.91 ± 12.03	65.95 ± 11.61	105.178	0.021	6.576	0.005
	post	69.27 ± 10.53 *	69.64 ± 10.53 ***	70.59 ± 10.84 ***	(<0.001)	(0.980)	(0.003)	0.003
Attention (s)	pre	22.55 ± 3.79	22.45 ± 3.79	25.14 ± 5.63	207.471	1.355	14.355	0.066
	post	21.23 ± 3.69 ***	21.18 ± 3.69 ***	22.45 ± 4.65 ***	(<0.001)	(0.265)	(<0.001)	0.019

All data represent mean ± standard deviation. CON, control group; STG, step-training group; RSTG, rhythm step training group. G, group; T, time; G × T, group × time interaction. As a result of paired *t*-test, if there is a significant difference between time, * means *p* < 0.05, and *** means *p* < 0.001.

## Data Availability

All the data are available in this paper.

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
