# Peer review of "Effects of Rhythm Step Training on Physical and Cognitive Functions in Adolescents: A Prospective Randomized Controlled Trial"

_healthcare, 2022, doi:10.3390/healthcare10040712_

Round 1

Reviewer 1 Report

According to the authors, the objective of this study was to evaluate the effectiveness of an exercise physical program with rhythm step training (RST) in ameliorating the cognitive and physical functions of female adolescents when compared to the control and step training groups. Interesting data were presented, and they can support the use of RST to improve both physical and cognitive functions in the adolescent population. However, it is necessary to clarify that some relevant information about the control group and step training group is missing. In addition, there are another couple of factors that should be concerned, too.

In the Abstract section:

1) Please describe if the control group was submitted to any exercise training or not.

2) Please clarify the meaning of this sentence "In terms of the difference and change in RST ability after 12 weeks, the RST score increased significantly in all groups, but the Δ% of the RSTG was higher than for the CON or STG." How was RST score increased in all groups since RST was only performed only by the RSTG?

3) Please provide the number of the percentages and p-value obtained in all parameters that presented significant differences.

4) Please join the following sentences "In addition, the verbal memory and attention of cognitive function were significantly improved in RSTG, comparing with the groups. Specifically, the Δ% of the RSTG increased more for verbal memory (7.52%) and attention (10.33%) than in the CG or STG."

In the Introduction section:

5) Although the COVID-19 pandemic imposed severe alteration in people's behavior, including reduction of physical activity, it is also noteworthy to add some pieces of information on the consequence of this situation in neurologic and cognitive aspects, particularly in the adolescent population.

In the Material and Methods section:

6) Please state if the control group was submitted to any program or not. All these volunteers did not perform any physical activities or exercises during the study period?

7) What was the step training program performed by the volunteers in the ST group? Again, they did not perform any other physical activities or exercises during the study period?

8) What was the period of the study? During the COVID-19 pandemic period?

9) Please move Table 1 from the Materials and Methods section to the Results section.

In the Results section

10) In relation to the physical characteristics of the participants, there were significant alterations in these parameters after the 12 weeks of exercise training or not in the volunteer groups?

11) In terms of the results presented in the subsection "3.2. Difference and Change of RST Ability" it is difficult to understand how the control group could increase RST scores without understanding what this group was submitted.

12) Please insert the values of the percentage for all results described in the text of the Results section since in some parts of this text the percentage is presented and in others not.

In the Discussion section:

13) Please revise the first paragraph of the Discussion section, because the abbreviation RST is used many times and difficult its comprehension.

14) The second paragraph presents information about the capacity of RST in enhancing physical and cognitive abilities. In my opinion, this paragraph could be useful to improve the understanding of the use of RST in the Introduction section. So, I suggest removing this paragraph from the Discussion section to the Introduction section.

15) Both in the Introduction and Discussion sections the authors mentioned the problem of obesity in children, but, in this study, they did not evaluate the effect of RST in obese or even overweight adolescents. Therefore, I suggest revising this topic and emphasizing the effect of sedentarism or low physical activities exclusively in the parameters evaluated in this study.

16) Please revise the limitations of the study since it was described is very succinct. I think that the authors could provide a better reflection about your study.

17) The authors mentioned in the Conclusion section that "Ultimately, RST can be used as a strategic tool to promote continuing voluntary participation in physical activity.", however, it was not presented any results concerning this subject in the present study. So, how the authors can support this proposal?

Author Response

Answers to 1st reviewer’s comments

Thank you for your kind advice and comments for publication in Healthcare. We revised our manuscript as per your comments. We represented the specific modifications in response to the comments by blue letters in our manuscript. We sincerely appreciate your comments because your comments make our manuscript better. Details of responses about reviewer’s comments are as follows:

In the Abstract section:

1) Please describe if the control group was submitted to any exercise training or not.

#Response 1: Thank you for your comments. Based on your comment, we have revised in the abstract as follows: “Sixty-six female middle-schoolers were divided into non-exercise group (control group, CON, n = 22), step training group (STG, n = 22), and rhythm step training group (RSTG, n = 22).”

2) Please clarify the meaning of this sentence "In terms of the difference and change in RST ability after 12 weeks, the RST score increased significantly in all groups, but the Δ% of the RSTG was higher than for the CON or STG." How was RST score increased in all groups since RST was only performed only by the RSTG?

#Response 2: The result description was incorrect. That is, in Table 4, the asterisk for the posterior value of CON was incorrectly entered. This was corrected and the contents of the abstract were also revised as follows.

“RST score increased significantly in the STG and RSTG compared to the CON after 12 weeks. Specifically, the Δ% of RSTG (11.44%) was higher than those of STG (9.01%) and CON (3.91%).”

3) Please provide the number of the percentages and p-value obtained in all parameters that presented significant differences.

#Response 3: Based on your comment, we provide the number of the percentages and p-value obtained in all parameters as follows: “After the end of experiment, the power (p < 0.001), agility (p < 0.001), muscle endurance (p < 0.001), dynamic or static balance (p < 0.001), and velocity (p < 0.001) of the RSTG were significantly improved than others. The power (2.36%), agility (13.34%), muscle endurance (5.08%), dynamic or static balance (3.50-9.18%), velocity (2.65%), and stride length (4.82%) of the RSTG Δ% were higher than those of the CON or STG. In addition, the verbal memory (p < 0.001) and attention (p < 0.001) of cognitive function were significantly improved in RSTG. Specifically, the Δ% of the RSTG increased more for verbal memory (7.52%) and attention (10.33%) than in the CON (verbal memory, 3.34%; attention, 5.83%) or STG (verbal memory, 5.85%; attention, 5.43%).”

4) Please join the following sentences "In addition, the verbal memory and attention of cognitive function were significantly improved in RSTG, comparing with the groups. Specifically, the Δ% of the RSTG increased more for verbal memory (7.52%) and attention (10.33%) than in the CG or STG."

#Response 4: Based on your comment, we have revised in the abstract as follows: “In addition, the verbal memory (p < 0.001) and attention (p < 0.001) of cognitive function were significantly improved in RSTG. Specifically, the Δ% of the RSTG increased more for verbal memory (7.52%) and attention (10.33%) than in the CON (verbal memory, 3.34%; attention, 5.83%) or STG (verbal memory, 5.85%; attention, 5.43%).”

In the Introduction section:

5) Although the COVID-19 pandemic imposed severe alteration in people's behavior, including reduction of physical activity, it is also noteworthy to add some pieces of information on the consequence of this situation in neurologic and cognitive aspects, particularly in the adolescent population.

#Response 5: Based on your comment, we have revised in the introduction as follows:

“According to one study, it was reported that adolescents are negatively affected not only in school life, but also in social life and outdoor activities due to the instability of COVID-19. Some of them have experienced an increase in domestic violence and report that similar events are directly affecting their psychological health, such as stress, anxiety and fear [2]. In addition, the decrease in social activities, such as restrictions on the use of sports facilities and school attendance due to COVID-19, is emerging as a big problem for adolescents in their growth stages [3,4].”

[References]

[2] de Figueiredo, C.S.; Sandre, P.C.; Portugal, L.C.L.; Mázala-de-Oliveira, T.; da Silva Chagas, L.; Raony, Í.; Ferreira, E.S.; Giestal-de-Araujo, E.; Dos Santos, A.A.; Bomfim, P.O. COVID-19 pandemic impact on children and adolescents' mental health: Biological, environmental, and social factors. Prog Neuropsychopharmacol Biol Psychiatry. 2021, 106, 110171. doi: 10.1016/j.pnpbp.2020.110171.

[3] Francesca, L.; Michele, D.C.; Milena, M.; Roberto, C. The impact of an extracurricular outdoor physical activity program on long-term memory in adolescent during COVID-19 pandemic. Journal of Human Sport and Exercise. 2021, 16, S1114-S1125. doi: 10.14198/jhse.2021.16.Proc3.28.

In the Material and Methods section:

6) Please state if the control group was submitted to any program or not. All these volunteers did not perform any physical activities or exercises during the study period?

#Response 6: Yes. The participants in the CON engaged neither in the STG nor in the RSTG. Based on your comments, we have supplemented on Line 120 as follows: “The participants in the CON engaged neither in the step training nor in the rhythm step training.”

7) What was the step training program performed by the volunteers in the ST group? Again, they did not perform any other physical activities or exercises during the study period?

#Response 7: The step training performed by the ST group is an exercise that is generally used in the past. It is to move the agility ladder (step ladder) quickly and repeatedly according to a set pattern without using music. To avoid the confusion, we inserted the sentence on Line 199 as follows. “For reference, the step training performed by the ST group is an exercise that is generally used in the past. It is to move the agility ladder (step ladder) quickly and repeatedly according to a set pattern without music.”

8) What was the period of the study? During the COVID-19 pandemic period?

#Response 8: Yes. This study was conducted from March 2020 to March 2021.

9) Please move Table 1 from the Materials and Methods section to the Results section.

#Response 9: Based on your comment, we have moved to the Results section.

In the Results section:

10) In relation to the physical characteristics of the participants, there were significant alterations in these parameters after the 12 weeks of exercise training or not in the volunteer groups?

#Response 10: Thank you for your kind advice and comments. We investigated and analyzed only the physical and cognitive functions relevant to the purpose of the study after 12 weeks. If there is an opportunity in the future, we will investigate changes in body composition after RST.

11) In terms of the results presented in the subsection "3.2. Difference and Change of RST Ability" it is difficult to understand how the control group could increase RST scores without understanding what this group was submitted.

#Response 11: In Table 4, the asterisk for the posterior value of CON was incorrectly entered. The asterisk symbol was removed, and the contents of the text were revised on Line 230. Meanwhile, although the control group did not participate in the exercise program (ST or RST) and only performed pre- and post-tests, their RST score was increased though not significantly. We think that an increase in the RST score in the control group caused by the learning effect of the pre-test. We hope you understand.

12) Please insert the values of the percentage for all results described in the text of the Results section since in some parts of this text the percentage is presented and in others not.

 #Response 12: Based on your comment, we have inserted the values of the percentage for all results and revised as follows:

” Specifically, the Δ% of the RSTG significantly was higher than that for the CON (3.91%), although there were no significant differences between the RSTG and STG (9.01%).”

“In terms of power, the RSTG increased significantly more than the CON (0.63%) or STG (1.11%). For agility, the RSTG increased significantly more than the CON (3.01%) or STG (5.70%), and the STG increased significantly more than the CON. In terms of muscular endurance, the RSTG increased significantly more than the CON (2.37%), and for dynamic balance ability, the RSTG (9.18%) and STG (8.20%) increased significantly more than the CON (1.52%).”

“Specifically, the anterior (3.50%) and anterior lateral (4.38%) lengths in the RSTG increased significantly more than those in the STG (2.60%, 2.74%). In terms of static balance, the anterior (2.60%) and lateral (3.96%) lengths in the STG increased significantly more than those in the CON (1.57%, 2.13%). For cadence (Figure 5), the CON (1.20%) increased significantly more than the RSTG (0.15%) or STG (0.60%), and the score for the STG was significantly higher than that for the RSTG. In terms of velocity, the RSTG (0.15%) increased significantly more than the STG (1.52%) or CON (1.15%).”

In the Discussion section:

13) Please revise the first paragraph of the Discussion section, because the abbreviation RST is used many times and difficult its comprehension.

#Response 13: Based on your comment, we have revised as follows: “This study found that the RST ability, power, agility, muscle endurance, balance ability, and gait ability of the rhythm step training group all significantly improved after the 12-week experiment was completed. And it was found that verbal memory and attention of RSTG were significantly improved after 12 weeks.”

14) The second paragraph presents information about the capacity of RST in enhancing physical and cognitive abilities. In my opinion, this paragraph could be useful to improve the understanding of the use of RST in the Introduction section. So, I suggest removing this paragraph from the Discussion section to the Introduction section.

#Response 14: Based on your comment, we have moved a part of paragraph to the Introduction section.

15) Both in the Introduction and Discussion sections the authors mentioned the problem of obesity in children, but, in this study, they did not evaluate the effect of RST in obese or even overweight adolescents. Therefore, I suggest revising this topic and emphasizing the effect of sedentarism or low physical activities exclusively in the parameters evaluated in this study.

#Response 15: Based on your comment, we have revised in the Introduction and Discussion sections.

16) Please revise the limitations of the study since it was described is very succinct. I think that the authors could provide a better reflection about your study.

#Response 16: Based on your comments, we have supplemented in the end of Discussion section as follows: ” However, this study has some limitations in that it is difficult to apply the findings to all age groups because the participants in this study were female adolescents. Furthermore, more extensive validation work with a larger sample size is needed in the future. Given these limitations, further studies investigating the effect on a more diverse and larger number of subjects are recommended.”

17) The authors mentioned in the Conclusion section that "Ultimately, RST can be used as a strategic tool to promote continuing voluntary participation in physical activity.", however, it was not presented any results concerning this subject in the present study. So, how the authors can support this proposal?

#Response 17: Based on your comment, we have revised in the conclusion as follows: “Ultimately, a rhythm step training can be used as a strategic tool to promote continuous voluntary participation in physical activities in school sports or social sports fields by utilizing appropriate music for the participants.”

We sincerely appreciate your comments again.

Reviewer 2 Report

The authors have tried to describe the Effects of Rhythm Step Training on Physical and Cognitive Functions in Adolescents: A Prospective Randomized
Controlled Tria. Manuscript you seeem curious in general. However, I have a few small comments that wil improve its quality:

  • Abstract : I suggest writing it with a breakdown (Background, Methods, Results, Conclusion)
  • Line 13 expand the shortcut 
  • Line 28 - limit the number of keyword 
  • 32-58- expand the introduction by doing a droader literature reviw
  • Maybe these items will be interesting?. I do not insist, but recommend:https://www.tandfonline.com/doi/abs/10.1080/13607863.2010.551341 https://journals.lww.com/nsca-jscr/Fulltext/2013/04000/Effect_of_Rhythm_on_the_Recovery_From_Intense.21.aspx   https://www.hindawi.com/journals/bmri/2020/2345607/  https://www.ncbi.nlm.nih.gov/pmc/articles/PMC5931151/  
  • The material and method are presented clearly and legibly
  • Table 1 - explain the abbreviations below the table (F, p) , two decimal places will be sufficient 
  • Table 2 - widen the second column 
  • Statistical analisys- I think it's wroth giving the size of the effect 
  • Results- The table is not very clean due to the large amount of data. Perhaps it is worthwhile to additionally emphasize the statistically significant values in bold
  • Add limitation of the study 
  • -Does you research have any practical application? It is wroth submitting an application form
  • -Declaration of informed consent - if the persons were under the age of majority, the consent of their parents and legal guardians should be mentioned 

Author Response

Author's Notes

Answers to 2nd reviewer’s comments

Thank you for your kind advice and comments for publication in Healthcare. We revised our manuscript as per your comments. We represented the specific modifications in response to the comments by blue letters in our manuscript. We sincerely appreciate your comments because your comments make our manuscript better. Details of responses about reviewer’s comments are as follows:

Abstract

#1. I suggest writing it with a breakdown (Background, Methods, Results, Conclusion)

#Response 1: Thank you for what the reviewer has pointed out the comment. We divided the abstract into Background, Materials and Methods, Results, and Conclusions according to the reviewers' comments as follows.

Abstract: Background: Rhythm step training (RST) for sensorimotor dual tasks is in the spotlight because it provides physical activity that is fun and allows participants to express various and creative movements, although it lacks a scientific evidence base. Therefore, this study was initiated to investigate how RST affects the physical and cognitive functions of adolescents. Materials and Methods: Sixty-six female middle-schoolers were divided into non-exercise group (control group, CON, n = 22), step training group (STG, n = 22), and rhythm step training group (RSTG, n = 22). To verify the combined effects of music-based rhythm and exercise, the program was conducted for 45 min/session a day, 3 times a week for 12 weeks. Results: RST score increased significantly in the STG and RSTG compared to the CON after 12 weeks. Specifically, the Δ% of RSTG (11.44%) was higher than those of STG (9.01%) and CON (3.91%). After the end of experiment, the power (p < 0.001), agility (p < 0.001), muscle endurance (p < 0.001), dynamic or static balance (p < 0.001), and velocity (p < 0.001) of the RSTG were significantly improved than others. The power (2.36%), agility (13.34%), muscle endurance (5.08%), dynamic or static balance (3.50-9.18%), velocity (2.65%), and stride length (4.82%) of the RSTG Δ% were higher than those of the CON or STG. In addition, the verbal memory (p < 0.001) and attention (p < 0.001) of cognitive function were significantly improved in RSTG. Specifically, the Δ% of the RSTG increased more for verbal memory (7.52%) and attention (10.33%) than in the CON (verbal memory, 3.34%; attention, 5.83%) or STG (verbal memory, 5.85%; attention, 5.43%). Conclusions: This study confirm that RST had a positive effect on the physical and cognitive functions of female middle-schoolers. We propose that rhythmic exercise combined with music is beneficial for adolescents’ physical and cognitive health.

#2. Line 13 expand the shortcut.

#Response 2: Thank you for what the reviewer has pointed out the comment. We corrected the Line 13 according to the reviewers' comments as follows.

“Therefore, this study was initiated to investigate how RST affects the physical and cognitive functions of adolescents.”

#3. Line 28 - limit the number of keyword

#Response 3: Thank you for what the reviewer has pointed out the comment. We corrected the Line 13 according to the reviewers' comments as follows. Response: Based on your comments, we reduced the Keywords as follows: adolescents; cognitive function; physical function; rhythm step training

#4. 32-58- expand the introduction by doing a droader literature reviw

Maybe these items will be interesting?. I do not insist, but recommend:

https://www.tandfonline.com/doi/abs/10.1080/13607863.2010.551341

https://journals.lww.com/nsca-jscr/Fulltext/2013/04000/Effect_of_Rhythm_on_the_Recovery_From_Intense.21.aspx

https://www.hindawi.com/journals/bmri/2020/2345607/ 

https://www.ncbi.nlm.nih.gov/pmc/articles/PMC5931151/

#Response 4: Thank you for what the reviewer has pointed out the comment. Based on your comment, we have supplemented in the introduction (Line 52, old version) as follows: “It is also reported that rhythmic exercise improved not only the health-related physical fitness level of children [Jo et al., 2018, Xu et al., 2020] and adults with intellectual disabilities, but also mobility and fear of falling in the elderly [Yamada et al., 2011].”

The material and method are presented clearly and legibly

#5. Table 1 - explain the abbreviations below the table (F, p) , two decimal places will be sufficient

#Response 5: Thank you for what the reviewer has pointed out the comment. Based on your comment, we explained F and p in the footnote.

Table 3. Physical characteristics of the participants.

Groups

CON (n = 22)

STG (n = 22)

RSTG (n = 22)

F

p

Age (y)

14.82 ± 0.80

14.68 ± 0.84

14.41 ± 0.96

2.652

0.078

Height (cm)

158.23 ± 5.44

158.55 ± 4.48

159.68 ± 4.29

0.567

0.570

Weight (kg)

60.36 ± 12.29

55.64 ± 9.6

54.73 ± 9.20

1.850

0.166

Percent fat (%)

24.14 ± 5.03

22.00 ± 3.52

21.45 ± 3.66

2.597

0.082

BMI (kg/m2)

20.60 ± 8.20

19.81 ± 5.92

23.31 ± 5.74

1.655

0.199

All data represent mean ± standard deviation. Comparative analysis was performed by one way ANOVA. BMI, body mass index; CON, control group; STG, step training group; RSTG, rhythm step training group. F and p are the results of one-way ANOVA for homogeneity verification.

#6. Table 2 - widen the second column

#Response 6: Based on your comments, we have revised Table 2 (old version).

#7. Statistical analisys- I think it's wroth giving the size of the effect

#Response 7: Based on your comments, we suggested the size of the effect. We inserted the context of effect size in the ‘Data Process and Statistical Analyses’ as follows. “Effect size (η2) was calculated according to Cohen’s d, which is equal to the mean difference of the groups divided by the pooled SD [Cohen, 1992].” In addition, the Effect size (η2) values represented in the ‘Results section’.

[Reference]

[33] Cohen, J. A power primer. Psychol. Bull. 1992, 112, 155-159. doi: 10.1037//0033-2909.112.1.155.

#8. Results- The table is not very clean due to the large amount of data. Perhaps it is worthwhile to additionally emphasize the statistically significant values in bold

#Response 8: We have revised Tables according to your comment.

#9. Add limitation of the study

#Response 9: We added the limitation of the study at the end of discussion according to your comment as follows.

” However, this study has some limitations in that it is difficult to apply the findings to all age groups because the participants in this study were female adolescents. Furthermore, more extensive validation work with a larger sample size is needed in the future. Given these limitations, further studies investigating the effect on a more diverse and larger number of subjects are recommended.”

#10. Does you research have any practical application? It is wroth submitting an application form

#Response 10: We don't understand what kind of suggestions the reviewers are making. In any case, we have a practical application, which has been approved by the Institutional Review Board of Chungnam National University (202001-SB-008-01).

Before the study, the principal investigator explained all procedures to the participants and their parents or legal guardians who then read and signed a consent form.

#11. Declaration of informed consent - if the persons were under the age of majority, the consent of their parents and legal guardians should be mentioned

#Response 11: Thank you for what the reviewer has pointed out the comment. Naturally, we obtained informed consent from the children's parents before the start of the experiment, and the IRB number was obtained from the school through the data. The content is inserted into the text on Line 95-98 as follows.

This study was conducted according to the Declaration of Helsinki (2013 version) and was approved by the Institutional Review Board of Chungnam National University (202001-SB-008-01).

We sincerely appreciate your comments again.

Round 2

Reviewer 1 Report

I would like to thank the authors for revising the manuscript based on the reviewers' comments, which improved its meaning and impact.
I have no more questions or comments.

Author Response

Dear Reviewer
The points you pointed out were of great help in revising and supplementing our thesis. Once again, thank you.